

# Estimates of direct radiative forcing due to aerosols from the MERRA-2 reanalysis over the Amazon region

Brunna Penna[1], Dirceu Herdies[1], and Simone Costa[1]

[1]National Institute for Space Research, SP, Brazil

**Correspondence:** B. R. Penna (brunna.romeropenna@nasa.gov)

**Abstract.** Sixteen years of analysis of clear-sky direct aerosol radiative forcing is presented for the Amazon region, with calcu-
lations of AERONET network, MODIS sensor and MERRA-2 reanalysis data. The results showed that MERRA-2 reanalysis
is an excellent tool for calculating and providing the spatial distribution of aerosol direct radiative forcing. In addition, the
difference between considering the reference state of the atmosphere without aerosol loading and with natural aerosol to obtain
the aerosol direct radiative forcing is discussed. During the dry season, the monthly average direct forcing at the top of atmo-
sphere varied from -9.60 to -4.20 W m$^{-2}$, and at the surface, it varied from -29.81 to -9.24 W m$^{-2}$, according to MERRA-2
reanalysis data and the reference state of atmosphere without aerosol loading. Already with the state of reference being the
natural aerosols, the average direct forcing at the top of atmosphere varied from -5.15 to -1.18 W m$^{-2}$, and at the surface, it
varied from -21.28 to -5.25 W m$^{-2}$; this difference was associated with the absorption of aerosols.

## 1  Introduction

The effect of aerosols on the climate system has been the subject of several scientific papers in recent years, due to the
increase in the concentration of anthropogenic aerosols; in addition, the effect of these constituents on the climate are not well
understood. The influence of aerosols on the Earth's energy balance occurs in two ways: i) the first interaction is through the
scattering and absorption of solar and infrared radiation, which alters the radiative balance of the earth-atmosphere system
and is called the direct effect (this is the focus of this work); ii) the second interaction is through the modification of the
microphysical properties of clouds (i.e., cloud albedo, cloud evolution, and precipitation efficiency) (Haywood and Boucher ,
2000), which is called the indirect effect.
Quantifying the influence of aerosols on the climatic system is based on the concept of radiative forcing. Radiative forcing
is defined by the difference in the net irradiance between a reference state and a disturbed state due to some climatic agent
(e.g., aerosols) (Foster et al., 2007). The reference state may be the total absence of atmospheric aerosols or the concentration
of aerosols during a given period (e.g., the wet season in the Amazon region) because the disturbed state corresponds to a large
concentration of aerosols in the atmosphere.
Aerosol direct radiative forcing (ARF), which is the focus of this paper, refers to a perturbation in the climatic system for
a given region or place in the world when considering only the direct effect of aerosols (i.e., the first interaction). The ARF
has been estimated by numerical models and satellite data, and the total amount of aerosols is generally obtained because





the ARF estimations by species of aerosols are less consistent. The latest IPCC report estimated a global ARF of magnitude
$0.35 \pm 0.5 \, \mathrm{W \, m^{-2}}$. This value indicates that aerosols continue to be major contributors of uncertainties when estimating and
interpreting climate behavior. This is mainly due to the short period of study and the numerical simplifications used when
estimating ARF. Commonly, these simplifications are associated with the representation of the distribution, composition and
radiative properties of the aerosols.
To obtain accurate aerosol direct radiative forcing, it is necessary to combine numerical models with observations over a long
period of time. An important tool that considers all of these items is reanalyses. Bellouin et al. (2013) were the first to estimate
the ARF from a reanalysis with aerosol data assimilation (the MACC reanalysis). The ARF estimation by Bellouin et al. (2013)
was performed for seven years (2000-2007) and found a global mean of $0.4 \pm 0.3 \, \mathrm{W \, m^{-2}}$, which reduced uncertainties in the
global ARF value. The aerosol data assimilated in the MACC reanalysis were only from the Moderate Resolution Imaging
Spectroradiometer sensor (MODIS). Now, the Global Modeling and Assimilation Office of the National Aeronautics and Space
Administration (GMAO-NASA) produced a second version of the MERRA reanalysis data, the MERRA-2, which extends to
the entire modern era of satellites (i.e., 1980 to present). This current reanalysis considers the modeling and observation of
atmospheric aerosols, which produces analysis fields for five species of aerosols. The assimilation of aerosols is performed by
coupling the Goddard Chemistry Aerosol Radiation and Transport (GOCART) chemical transport model and the MERRA-2
reanalysis radiation data. The assimilated aerosol data comes from several instruments, such as the Aerosol Robotic Network
(AERONET), the MODIS sensor, and the Advanced Very High-Resolution Radiometer (AVHRR). Thus, MERRA-2 is a great
tool for studies on aerosols and their impacts on climate.
The value of the global ARF does not specify the extent that certain species of aerosol impacts the global climate (for exam-
ple, the impact of aerosols from biomass burning on the Amazon). Thus, to better understand ARF values in the climate system,
assessments for specific locations are indispensable. Procopio et al. (2004) estimated the ARF over the Amazon region during
the dry season and found values of $-39.5 \pm 4.2 \, \mathrm{W \, m^{-2}}$ on the surface and $-8.3 \pm 0.6 \, \mathrm{W \, m^{-2}}$ at the top of the atmosphere.
These values were obtained using aerosol optical depth (AOD) from the AERONET network. Liu (2005) and Patadia et al.
(2008) estimated the ARF (also over Amazon region) as $-16.5 \, \mathrm{W \, m^{-2}}$ and $-13 \, \mathrm{W \, m^{-2}}$ at the surface and the top of the atmo-
sphere, respectively. In the South American region, Zhang et al. (2008) found ARF values between $-35 \, \mathrm{W \, m^{-2}}$ and $-10 \, \mathrm{W \, m^{-2}}$
at the surface and $-8 \, \mathrm{W \, m^{-2}}$ and $-1 \, \mathrm{W \, m^{-2}}$ at the top of the atmosphere. All of these estimates were obtained under clear-sky
conditions and made with satellite data or climate models.
The goal of this work is to obtain the estimation of clear-sky ARF values at the surface, atmosphere and top of the atmosphere
during a 16-year period (2000-2015) over the Amazon region using data from the MERRA-2 reanalysis. This estimation is
obtained using two methods found in the literature and from the AERONET and MODIS data. In this context, it is possible to
evaluate the interannual variability of ARF over the Amazon region and discuss the methods of obtaining ARF.
The dataset and the study region are described in section 2. Section 3 shows the concepts and methodologies when estimating
ARF. The analysis of AOD values for the study period is presented in section 4. The ARF results and discussions, as well as
the conclusions, are given in sections 5 and 6, respectively.





**2 Data and Region of Study**
The study area chosen to estimate ARF was the Amazon. The climate in this region is the result of the combination of several
factors, where the availability of solar energy is the most important. The greatest amount of total surface radiation occurs during
the dry season (DS), which coincides with the occurrence of large-scale annual burnings. Biomass burning affects not only the
Amazon, but it is also exported to the southeastern part of the country and neighboring countries, such as Bolivia, depending
on the amount of particulate matter emitted during these times (Freitas et al., 2005). The Amazon area (AM) used is the same
as that in Procopio et al. (2004), which is defined between 0°- 20° S and 45° W - 65° W. The ARF estimation was also obtained
in the Alta Floresta (AF; 9° 55' S - 56° W), this site has an AERONET tower with good quality data for the entire study period.
A brief description of the sensors and data used is given below. Further details on the data are available in the literature based
on the AERONET (Holben et al., 1998), MODIS (Remer et al., 2005) and MERRA-2 reanalysis (Gelaro et al., 2017).
**2.1 AERONET**
AERONET is a federation of well-calibrated ground-based sun photometers; these instruments provide a global estimate of
the aerosol optical depth at seven wavelengths, along with other inversion products (Holben et al., 1998). The uncertainty in
AOD estimation is approximately $\pm$ 0.15 (Holben et al., 2001). We use the monthly average of the AOD at wavelengths of
440, 500 and 670 nm for 16 years at the Alta Floresta station. The level of quality for these data is level 2, version 2. The AOD
of 550 nm (same as that in MERRA-2) was obtained through the mathematic expression below:

$$\frac{AOD_\lambda}{AOD_{\lambda_0}} = (\frac{\lambda}{\lambda_0})^{-\alpha} \tag{1}$$

where, $\lambda_0$ represents the desired wavelength (550 nm), $\lambda$ represents the wavelength at 500 nm (close to the desired one) and
$\alpha$ represents the Angström coefficient, which is given by:

$$\alpha = \frac{log\frac{AOD_{\lambda_1}}{AOD_{\lambda_2}}}{log\frac{\lambda_1}{\lambda_2}} \tag{2}$$

where, $\lambda_1$ and $\lambda_2$ are the wavelengths at 670 nm and 440 nm, respectively.
**2.2 MODIS**
The MODIS sensor aboard both the Terra and Aqua NASA satellites records near-global daily observations of the Earth across
a wide spectral range (0.41 - 15 m), with a swath width of 2330 km. These measurements are used to derive spectral aerosol
optical depths and aerosol size parameters over land and ocean surfaces. Over the continents, the estimated uncertainty is
approximately $\pm 0.05 \pm 0.15 \times$ AOD (at 550 nm) (Remer et al., 2005). The dataset used in this work belonged to collection
6, which had a resolution of 3 km and a quality control level of 2. The files are named MOD04_3K and MYD04_3K for the



satellites Terra and Aqua, respectively. The AOD data for the Terra satellite were used from January 2000 and from July 2002
until December 2015 the AOD data for the Aqua satellite also were included.

## 2.3 MERRA-2 aerosol reanalysis

Briefly, the Modern-Era Retrospective Analysis for Research and Application - version 2, as described in Gelaro et al. (2017),
uses the Goddard Earth Observing System-5 (GEOS-5) atmospheric general circulation model (Molod et al., 2015) and the
three-dimensional variational data assimilation (3DVar) Gridpoint Statistical Interpolation analysis system (GSI; Wu et al.
(2002)). The GEOS-5 model is radiatively coupled with the GOCART model (Chin et al., 2002) and simulates five species
of aerosols (i.e., dust, sea salt, black carbon, organic carbon and sulfate). In the MERRA-2 reanalysis, the meteorological
and aerosol observations are simultaneously assimilated within the GEOS-5. The aerosol optical depth observations that are
assimilated come from the MODIS Neural Net Retrieval and AVHRR Neural Net Retrieval at 550 nm (Randles et al. , 2017).
In addition, observations from the space-based Multiangle Imaging SpectroRadiometer (MISR) for bright surfaces (i.e., albedo
> 0.15) and observations from the AERONET network are also assimilated at 550 nm. The AOD measurements from the
AERONET network are interpolated to 550 nm using the Angström ratio. The model resolution is $0.5° \times 0.625°$ latitude by
longitude, respectively, with 72 hybrid-eta layers from the surface through 0.01 hPa. The assimilation of aerosols is performed
every 3 hours. The total AOD represents extinction in the atmospheric column due to presence of all species of aerosols. More
details on the optical properties of aerosols in MERRA-2 can be found in Colarco et al. (2010).

## 3 Methodology

Clear-sky aerosol direct radiative forcing was obtained by two methods over the Amazon region and the Alta Floresta, during
the years 2000 to 2015. The first method used to obtain ARF values under clear-sky conditions was the parameterization
obtained by Procopio et al. (2004). This method estimated ARF at Earth's surface (SUR), throughout the atmosphere (ATM)
and at the top of the atmosphere (TOA) over vegetated areas, such as the Amazon region. This parameterization, called method
1 (M1) here, is a function only of the aerosol optical depth, which is expressed as:

$$ARF1_{SUR} = 5.04(AOD)^2 - 51.6(AOD) + 3.92 \qquad (3)$$

$$ARF1_{TOA} = -0.95(AOD)^3 + 6.71(AOD)^2 - 16.5(AOD) + 1.57 \qquad (4)$$

The ARF integrated over the atmosphere was obtained by subtracting the ARF at the top of the atmosphere from the ARF
at the surface. The uncertainty in the estimate was $\pm 4\%$ and $\pm 6\%$ for an AOD=1 at the surface and top of the atmosphere,
respectively. The reference AOD in this method represented the transition between the wet season and the dry season in the
Amazon region and had a value equal to 0.11 (Procopio et al., 2004). For ARF estimation, the AOD was the monthly average
aerosol optical depth from the datasets described above.



The second method, which is most commonly used by the scientific community, is defined by the difference between the
liquid solar flux (i.e., the difference between the descending and ascending radiation) in the presence ($RN_{aero}$) and absence
($RN_{noaero}$) of atmospheric aerosols, where the absence of aerosols results in an AOD=0. To calculate the ARF by this method,
the radiative variables of the MERRA-2 reanalysis were used. Method 2 (M2) was also calculated under clear-sky conditions
and the monthly average of variables SWGNTCLR (surface net downward shortwave flux assuming clear-sky), SWGNT-
CLRCLN (surface net downward shortwave flux assuming clear-sky and no aerosol), LWGNTCLR (surface net downward
longwave flux assuming clear sky) and LWGNTCLRCLN (surface net downward longwave flux assuming clear-sky and no
aerosol) are considered to estimate the ARF at surface. In the top of atmosphere, the variables considered are SWTNTCLR (toa
net downward shortwave flux assuming clear sky), SWTNTCLRCLN (toa net downward shortwave flux assuming clear-sky
and no aerosol), LWTUPCLR (upwelling longwave flux at toa assuming clear-sky) and LWTUPCLRCLN (upwelling long-
wave flux at toa assuming clear-sky and no aerosol), located on the file tavgM_2d_rad_Nx. The ARF at atmosphere can be
obtained subtracting ARF at the TOA from ARF at surface. The method 2 can be expressed by:

$$ARF2 = RN_{aero} - RN_{noaero} \tag{5}$$

or,

$$ARF2_{SUR} = (SWGNTCLR + LWGNTCLR) - (SWGNTCLRCLN + LWGNTCLRCLN) \tag{6}$$

$$ARF2_{TOA} = (SWTNTCLR + LWTUPCLR) - (SWTNTCLRCLN + LWTUPCLRCLN) \tag{7}$$

**4   Evaluation of AOD from the MERRA-2 aerosol reanalysis**
To calculate the ARF by method 1, aerosol optical depths from observations of the AERONET network, MODIS sensor and
MERRA-2 reanalysis are considered. Figure 1 shows the AOD monthly average for the period 2000 to 2015 from the MERRA-
2 reanalysis and the AERONET network near the Alta Floresta. The AOD values of these two datasets show good agreement,
presenting greater differences only in the months when the AOD values are higher than 0.25 (i.e., during the dry season in
the Amazon region). During this same period, higher standard deviation values are also verified. In general, data from the
MERRA-2 reanalysis underestimates the AOD values observed by the AERONET network during the burning period in South
America. The same is observed when we compare the AOD data from the MERRA-2 reanalysis and the AOD data from the
MODIS sensor onboard the TERRA and AQUA satellites, which is discussed below.
In Figure 1, it is observed that the months of April, May and June present the lowest values of AOD, while the month of
September presents the highest values of AOD near the Alta Floresta. In September, during the period from 2000 to 2015, the



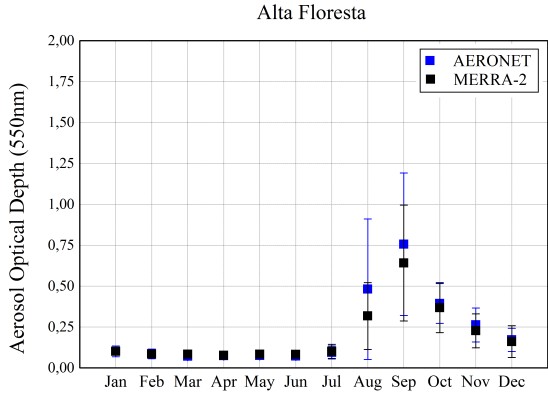

**Figure 1.** AOD monthly average and standard deviation for the period from 2000 to 2015 at the Alta Floresta location for the AERONET network data (in blue) and for the MERRA-2 reanalysis data (in black).

AOD monthly average is $0.64 \pm 0.35$ and $0.76 \pm 0.44$ for the MERRA-2 reanalysis and AERONET, respectively. These values
expose the excessive human activity of biomass burning and the large insertion of aerosols into the atmosphere.

3       The AOD averages obtained from the MERRA-2 reanalysis, the AERONET network and the MODIS sensor for the dry

season from 2000 to 2015 can be found in table 1. For the MERRA-2 reanalysis, AOD values are obtained over a specific
location, such as the AERONET network station located near the Alta Floresta; AOD values are also averaged over Amazon
region. The AOD climatological average observed at the Alta Floresta during the dry season is $0.54 \pm 0.29$ and $0.44 \pm 0.17$
when estimated by the reanalysis. The AOD climatological average in the Amazon region, estimated by the MODIS sensor,
is $0.39 \pm 0.15$ and $0.29 \pm 0.08$ when obtained from MERRA-2 reanalysis. Note that the AOD from the reanalysis tends to
be smaller when compared to that of the remote sensing data. The MERRA-2 reanalysis shows a better agreement with the
MODIS sensor data ($R^2 = 0.919$) than with the AERONET network ($R^2 = 0.888$). This is understandable because there is a
much larger number of observations from the MODIS sensor that are assimilated in the MERRA-2 reanalysis than there are
observations from the AERONET station (near the Alta Floresta). These three data sources show that 2002, 2005, 2007 and
2010 presented months with the highest aerosol loads during the dry season, while 2009 and 2013 presented the lowest aerosol
values (Table 1).

15       The interannual variability of the AOD, for the period 2000 to 2015, is shown in Figure 2. The highest aerosol loads on

in the Amazon region, as discussed above, were in the years of 2002 ($AOD_{MODIS} = 0.64$ vs. $AOD_{MERRA-2} = 0.44$),
2005 ($AOD_{MODIS} = 0.73$ vs. $AOD_{MERRA-2} = 0.51$), 2007 ($AOD_{MODIS} = 0.95$ vs. $AOD_{MERRA-2} = 0.72$) and 2010
($AOD_{MODIS} = 0.80$ vs. $AOD_{MERRA-2} = 0.59$), in the month of September. In During these last three years, an increase
was also observed in the number of fires and negative precipitation anomalies, which that reached -15.72 mm/h (2005), -
34.72 mm/h (2007) and -29.72 mm/h (2010) in the region (Prado and da Costa Coelho , 2017). However, 2009 and 2013



**Figure 2.** A sixteen year analysis of aerosol optical depth (550 nm) over the Amazon region. Data from MODIS sensor are represented by the solid red line, and data from the MERRA-2 reanalysis are represented by the solid black line.

**Table 1.** Dry season average aerosol optical depths (August to October) between 2000-2015.

| Year | Alta Floresta | | Amazon Region | |
| | MERRA-2 | AERONET | MERRA-2 | MODIS |
|---|---|---|---|---|
| 2000 | 0.34 | 0.40 | 0.22 | 0.35 |
| 2001 | 0.42 | 0.45 | 0.27 | 0.37 |
| **2002** | **0.66** | **0.94** | **0.35** | **0.53** |
| 2003 | 0.40 | 0.50 | 0.26 | 0.38 |
| 2004 | 0.55 | 0.64 | 0.32 | 0.54 |
| **2005** | **0.65** | **1.02** | **0.42** | **0.58** |
| 2006 | 0.47 | 0.59 | 0.28 | 0.40 |
| **2007** | **0.96** | **0.92** | **0.52** | **0.69** |
| 2008 | 0.35 | 0.37 | 0.26 | 0.31 |
| 2009 | 0.19 | 0.20 | 0.18 | 0.18 |
| **2010** | **0.74** | **0.99** | **0.43** | **0.58** |
| 2011 | 0.25 | 0.28 | 0.21 | 0.23 |
| 2012 | 0.31 | 0.38 | 0.27 | 0.31 |
| 2013 | 0.19 | 0.19 | 0.17 | 0.20 |
| 2014 | 0.25 | 0.28 | 0.22 | 0.28 |
| 2015 | 0.36 | - | 0.30 | 0.34 |
| **Mean** | **0.44** | **0.54** | **0.29** | **0.39** |



presented the lowest aerosol optical depths over the Amazon, being where the year 2009 with had the lower weakest intensity
of fires and a positive precipitation anomaly.
**5   Clear-sky aerosol direct radiative forcing results**
To obtain a long and accurate time series of the ARF in the Amazon region, the MERRA-2 reanalysis data were used. This tool
assimilates the aerosol optical depth with several instruments, including data from the AERONET network and the MODIS
sensor. ARF estimation was also obtained through the observed values of the AERONET network (near the Alta Floresta sta-
tion) and the MODIS sensor. Figure 3a shows the ARF temporal distribution near the Alta Floresta, which was calculated from
the AOD data from the MERRA-2 reanalysis and method 1. In this figure, it can be noted that the aerosol direct radiative forcing
had a greater magnitude in 2007 and 2010 in the month of September, reaching values of -62.62 W m$^{-2}$ and -48.71 W m$^{-2}$,
respectively, at the surface; this was expected because method 1 is proportional to the aerosol optical depth. On the other hand,
in September of 2009, the ARF at the surface was found to be -7.04 W m$^{-2}$. During a period of 16 years (2000-2015), when
only analyzing the month of September, the ARF at the surface ranged from -62.62 (2007) to -7.04 W m$^{-2}$ (2009) and from
-11.33 (2007) to -1.70 W m$^{-2}$ (2009) at the top of the atmosphere, where the differences were associated with the absorption
of aerosols throughout the atmosphere at magnitudes of 5.33 (2007) to 51.30 W m$^{-2}$ (2009). The same calculation was made
with data from the AERONET network (near the Alta Floresta), and some results for the dry season can be seen in Table 2.

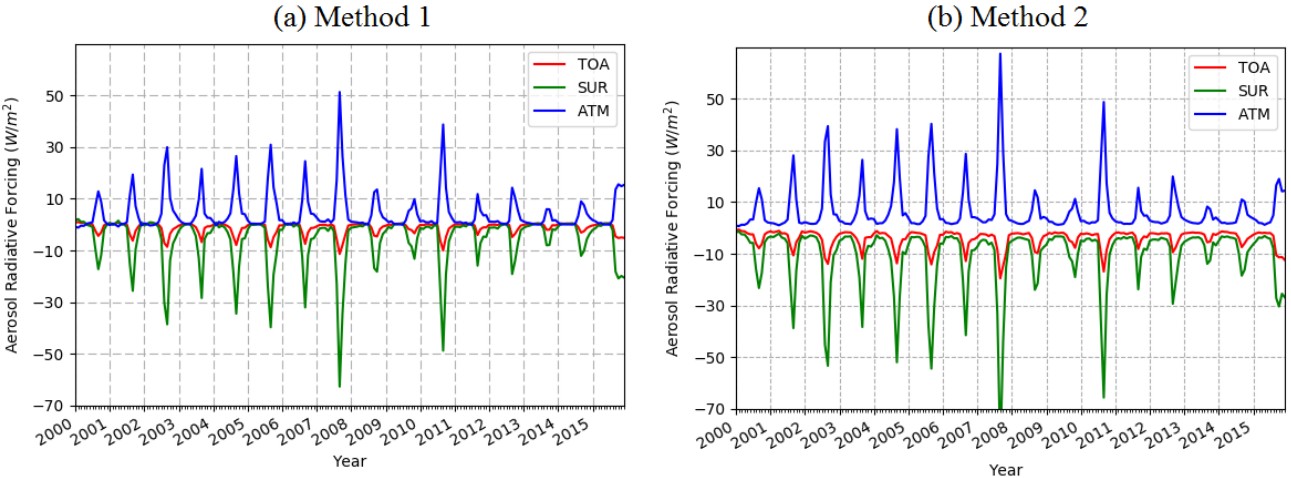

**Figure 3.** Time series of the ARF (in W m$^{-2}$) at the Alta Floresta, which were calculated from the MERRA-2 reanalysis data and a) Method
1 and b) Method 2. Aerosol radiative forcing is represented at the top of the atmosphere (solid red line), throughout the atmosphere (solid
blue line) and at the surface (solid green line).





**Table 2.** Dry season averages of aerosol direct radiative forcing, in $W\,m^{-2}$, at the surface (SUR), at the top of the atmosphere (TOA) and throughout the atmosphere (ATM) at the Alta Floresta and for the Amazon region during the period from 2000 to 2015.

| Data | Method | AOD | $ARF_{SUR}$ | $ARF_{ATM}$ | $ARF_{TOA}$ |
|---|---|---|---|---|---|
| **Alta Floresta** | | | | | |
| AERONET | 1 | 0.54±0.21 | -21.90±8.31 | 17.01±6.74 | -4.89±1.59 |
| MERA-2 | 1 | 0.44±0.17 | -17.51±7.89 | 13.41±6.24 | -4.09±1.66 |
| MERRA-2 | 2 | - | -25.97±10.01 | 17.83±7.56 | -8.14±2.59 |
| **Amazon Region** | | | | | |
| MODIS | 1 | 0.39±0.11 | -15.40±5.02 | 11.65±3.84 | -3.75±1.12 |
| MERRA-2 | 1 | 0.29±0.08 | -10.66±4.09 | 8.05±3.03 | -2.62±1.06 |
| MERRA-2 | 2 | - | -16.49±4.66 | 10.31±3.38 | -6.18±1.48 |

Figure 3b shows the temporal distribution of ARF near the Alta Floresta, which was calculated with the MERRA-2 reanalysis
data and method 2. From this method, the ARF at the surface for the month of September varied from -86.90 (2007) to
-11.18 $W\,m^{-2}$ (2009), from -19.47 (2007) to -4.86 $W\,m^{-2}$ (2009) at the top of the atmosphere and from 6.32 (2007) to
67.43 $W\,m^{-2}$ (2009) throughout the atmosphere.
Near the Alta Floresta, the estimation of ARF obtained from the observed data of the AERONET network had a better
agreement with the ARF estimated by the MERRA-2 reanalysis and method 1 (M2M1). Over the Amazon region, the ARF
estimation from the MODIS sensor data had a better agreement with the ARF estimation by the MERRA-2 reanalysis and
method 2 (M2M2) both at the surface and throughout the atmosphere. At the top of the atmosphere, the values of ARF obtained
with MODIS had a better agreement with the MERRA-2 reanalysis and method 1 (M2M1). For both the regions near the Alta
Floresta and for the entire Amazon, the greatest differences in the ARF values between methods 1 and 2 were at the top of the
atmosphere. Figure 4 shows the values found for ARF at the top of the atmosphere over the Amazon region.
It should be noted that in Figure 3a, there is a greater proximity between the curves representing the ARF at the surface and
throughout the atmosphere compared to that in Figure 3b. This is explained by the fact that method 1 considers an atmosphere
without aerosols equal compared to that under natural conditions (i.e., the Amazon region during the rainy season). Under this
condition, the aerosol optical depth is equal to 0.11, and the aerosol radiative forcing for this value is adjusted to be very close
to zero. In method 2, an aerosol-free atmosphere indicates an aerosol optical depth equal to zero (AOD=0). Thus, in method
1, when the AOD=0.11, the estimated ARF is approximately -1.7 $W\,m^{-2}$ at the surface and -0.23 $W\,m^{-2}$ at the top of the
atmosphere (i.e., very close to zero). In method 2, when the AOD=0.11, the ARF values are approximately -5.74 $W\,m^{-2}$ at the
surface and -2.67 $W\,m^{-2}$ at the top of the atmosphere. The correlation between the aerosol optical depth from the MERRA-2
reanalysis and ARF estimated by the two methods can be seen in Figure 5. It can be seen that the correlation between the AOD
and ARF values obtained by method 1 are better than when the AOD and the ARF are compared and obtained by method 2.



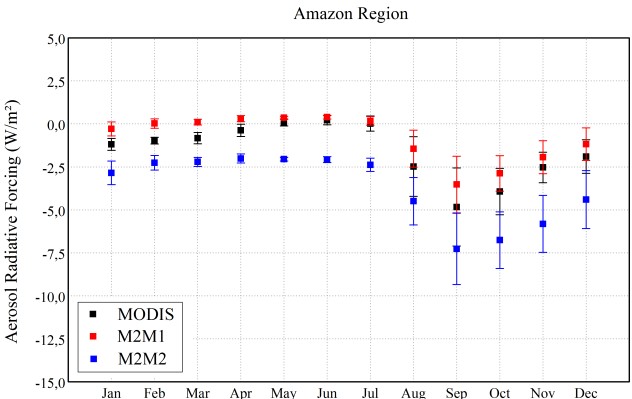

**Figure 4.** Averages of aerosol direct radiative forcing for the period 2000 to 2015 at the top of the atmosphere over the Amazon region. Black represents the ARF obtained through the MODIS sensor data by method 1, red represents the ARF obtained through the MERRA-2 reanalysis data by method 1 (M2M1) and the color blue represents the ARF obtained through the MERRA-2 reanalysis data by method 2 (M2M2).

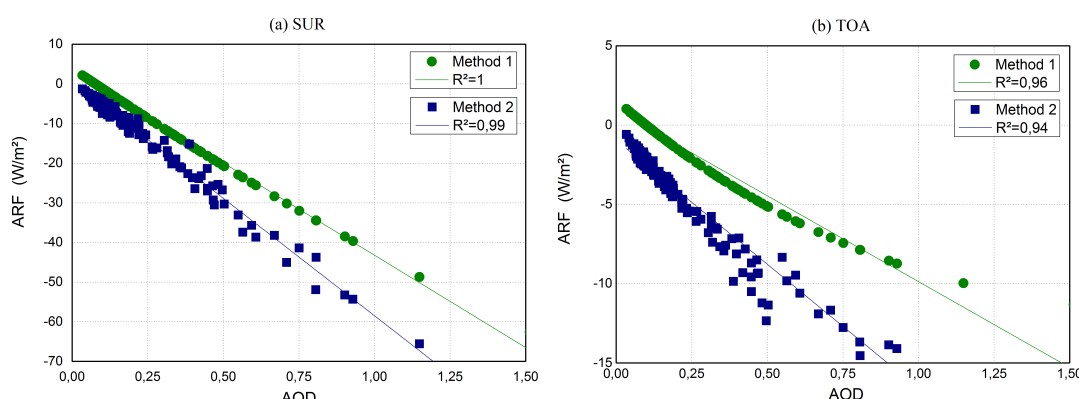

**Figure 5.** Relation between the aerosol optical depth and the aerosol direct radiative forcing estimated, at the Alta Floresta, with the MERRA-2 reanalysis data at the surface (a) and at the TOA (b) by method 1 (in solid circles) and method 2 (in solid squares). The straight line represents the linear fit for both methods.

1    It is also observed that the correlation between the AOD and the ARF for both methods is better at the surface. According to
2    Sena et al. (2013), the linearity between ARF at the top of the atmosphere and the AOD is not expected for high AOD values.





The monthly average for aerosol direct radiative forcing during the period from 2000 to 2015 was also calculated for the
Amazon region. The results found for the Amazon region were smoother than those at a specific point in the Amazon, such as
near the Alta Floresta. This is because the average was calculating when considering all of the grid points located inside this
large region, and this involved locations where the ARF was very small. Calculations for the ARF over the Amazon region were
performed using the two methods described above and data from the MERRA-2 reanalysis and the MODIS sensor onboard the
Aqua and Terra satellites. The average values during the dry season are also shown in Table 2. The results from both methods
presented in this paper agree with previous studies, such as Procopio et al. (2004), who found values between -74 to 21 $\mathrm{W\,m^{-2}}$
for ARF at the surface and -12 to -5 $\mathrm{W\,m^{-2}}$ at the TOA at the Alta Floresta, and Zhang et al. (2008) found values of the ARF
from -35 to -10 $\mathrm{W\,m^{-2}}$ at the surface and -8 to -1 $\mathrm{W\,m^{-2}}$ at the TOA over South America (Table 3).

**Table 3.** Comparisons of clear-sky aerosol direct radiative forcing, in $\mathrm{W\,m^{-2}}$, with previous studies.

| Region | Period | Data | AOD | $ARF_{SUR}$ | $ARF_{ATM}$ | $ARF_{TOA}$ | Reference (Condition) |
|---|---|---|---|---|---|---|---|
| Alta Floresta | DS (1993-2002) | AERONET | 0.55 to 1.29 | -22.9 to -54.3 | 17.3 to 43.7 | -5.6 to -10.6 | Procopio et al. (2004) (24hi) |
| Abracos Hill | DS (1994-2002) | | 0.52 to 1.83 | -21.5 to -73.6 | 16.2 to 61.7 | -5.3 to -12 | |
| AM | Jul-Dec (2002) | MODIS | 0.29 to 0.59 | -10.6 to -24.8 | - | -2.7 to -6 | |
| AM | Aug-Dec (2000-2005) | MISR | 0.24 ± 0.8 | - | - | -7.6 ±1.9 | Patadia et al. (2008) (24h) |
| AM | Aug-Sep (2000-2009) | CERES | 0.25±0.11 | - | - | -5.6±1.7 | Sena et al. (2013) (24h) |
| South America | Aug-Sep (2002) | MODIS Climate Model | - | - | - | -8 to -1 | Zhang et al. (2008) (24h) |
| South America | 2000-2005 | CERES MODIS | - | - | - | -1 to -0.2 | Quass et al. (2008) (Annually) |

*24h means daily average

*i Atmosphere background condition for AOD is 0.11

DS means Dry Season



**5.1    Spatial distribution of clear-sky aerosol direct radiative forcing**
Figure 6 shows the seasonal distribution of aerosol direct radiative forcing over South America obtained with the MERRA-2
reanalysis during 2000-2015. In terms of the magnitude of the aerosol direct radiative forcing at the surface, the scenario is
dominated by aerosols due to biomass burning over the central region of the continent during July-August-September (JAS, $\simeq$
12.77 W m$^{-2}$), followed by October-November-December (OND, $\simeq$ 11.48 W m$^{-2}$), where the largest magnitude of ARF at
surface is identified in the northeastern portion of the Amazon basin, which is also due to biomass burning. The OND season
is less intense because the area of fires is smaller and better ventilated, which favors the dispersion of aerosols in this region.
These months extend throughout the dry season over the Amazon region and present large areas that are dominated by ARF at
the surface, with values less than -20 W m$^{-2}$ and -15 W m$^{-2}$ for JAS and OND, respectively. The months of January-February-
March (JFM) present a small area with ARF at the surface, with values between -10 and -12.5 W m$^{-2}$ in the region including
Venezuela, Colombia, Ecuador and northern Peru, which is also due to biomass burning. The season of JFM, as well as April-
May-June (AMJ), results in ARF values at surface of approximately -12.5 W m$^{-2}$ in the north and northeastern regions of
South America, along the oceanic coastal region. According to Rosário (2011), this region is dominated by an aerosol plume
from the Saharan desert, which contributes to these ARF values.

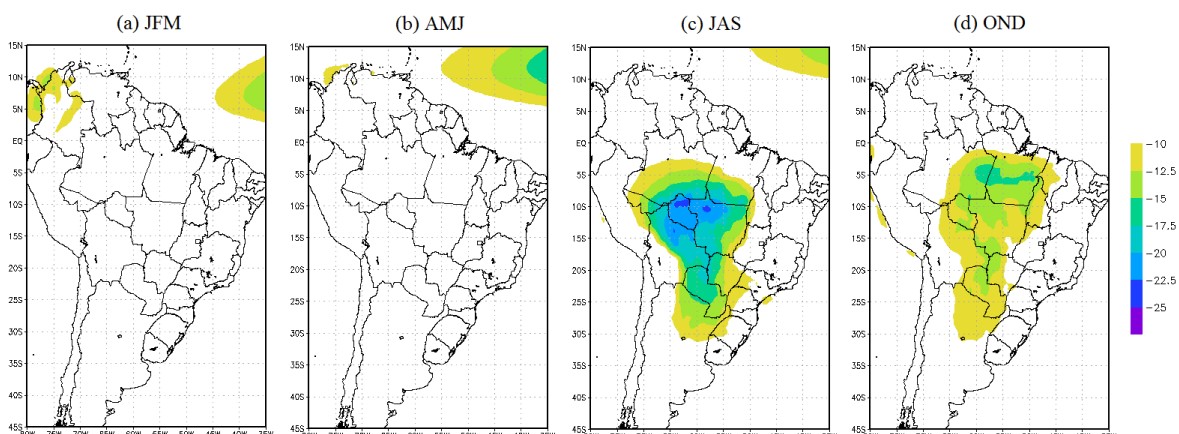

**Figure 6.** The seasonal distribution of aerosol direct radiative forcing at the surface (in W m$^{-2}$) over the region of South America from
MERRA-2 reanalysis data and method 2 during 2000-2015 in the months of a) JFM, (b) AMJ, (c) JAS and (d) OND.

15       In the period analyzed, the years of 2005, 2007 and 2010 presented the largest areas of ARF over South America, with

magnitudes less than -40 W m$^{-2}$ at the surface and -10 W m$^{-2}$ at the top of the atmosphere during the dry season. Figure 7a
shows the climatological distribution of ARF at the surface for the month of September, and it was noted that in 2007 (Figure
7b), the values of ARF at the surface were extremely lower than those in the climatology for a large portion of the region.
The ARF anomaly at the surface for the year 2007 had magnitudes ranging from -40 W m$^{-2}$ to -10 W m$^{-2}$ (Figure 8a). This



anomaly indicated that the magnitudes were comparable to the magnitude of ARF at the surface. In 2009, there was a large-
scale precipitation anomaly in the Brazilian Amazon (Vale et al., 2011), and the values of ARF at the surface (Figure 7c) and
the top of the atmosphere were not below -20 W m$^{-2}$ and -10 W m$^{-2}$, respectively. The ARF surface anomaly for the year
2009 is shown in Figure 8b.

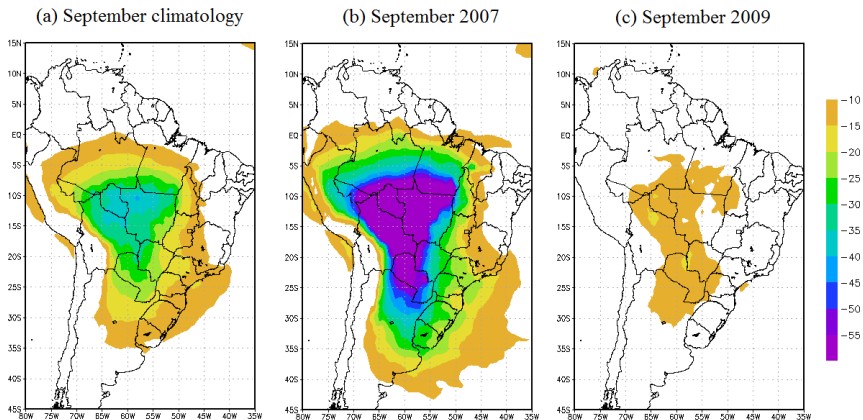

**Figure 7.** The spatial distribution of aerosol direct radiative forcing at the surface (in W m$^{-2}$) obtained from the MERRA-2 reanalysis data
and method 2 for the a) September climatology (2000-2015), b) September of 2007 and c) September of 2009.

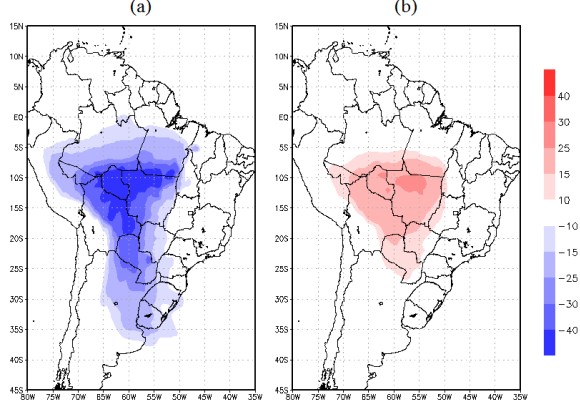

**Figure 8.** Anomalous aerosol direct radiative forcing at the surface (in W m$^{-2}$) from the MERRA-2 reanalysis for 2007 (a) and 2009 (b).




# 6 Conclusions

The estimation of aerosol direct radiative forcing from the MERRA-2 reanalysis for the Amazon region is of great importance for the understanding of how aerosols affect the regional and even global climate because the data of the MERRA-2 reanalysis include strengths of the observations and models. The observations provide a current estimate of the aerosol distribution, while the numerical model provides estimates with temporally and spatially homogeneous coverages.

The average aerosol optical depth from 2000 to 2015 for the dry season in the Amazon region was $0.29 \pm 0.08$, reaching a value of $0.44 \pm 0.17$ for a specific location, such as the Alta Floresta. The ARF at the top of the atmosphere was $-2.62 \pm 1.06\,\mathrm{W\,m^{-2}}$ and $-6.18 \pm 1.48\,\mathrm{W\,m^{-2}}$ by Methods 1 and 2, respectively. At the surface, the ARF reached values of $-10.66 \pm 4.09\,\mathrm{W\,m^{-2}}$ with method 1 and $-16.49 \pm 4.66\,\mathrm{W\,m^{-2}}$ with method 2. An important difference between the methods is the aerosol optical depth of the background atmosphere, where method 1 considered an AOD value equal to 0.11 and method 2 considered an AOD value equal to zero. This concept is one of the reasons that these two methods show several differences in ARF estimates at the top of the atmosphere. The values of the ARF at the surface and the top of the atmosphere agree with previous works.

ARF estimates from the MERRA-2 reanalysis aerosol product provide several applications. First, these surface ARF values can be used to adjust the flows that reach the Earth's surface, allowing for the better estimation of sensible and latent heat fluxes. Thus, a better estimation of precipitation can be obtained because a smaller surface incident flux favors a decrease in evaporation. The spatial distribution of ARF allows for the understanding of temporal and spatial patterns of ARF magnitudes over a given region and location. In addition, the MERRA-2 reanalysis products can be used as initial conditions for seasonal and decadal numerical forecasting, which require the knowledge of contour conditions in the climate system. Finally, MERRA-2 reanalysis products can help to measure the impact of aerosols on the climate.

This was the first study to present ARF estimates at the surface, throughout the atmosphere and at the top of the atmosphere over the Amazon region for a long period of time by analyzing the interannual variability of aerosol optical depth and aerosol direct radiative forcing over a region that is strongly impacted by aerosols during the dry season.

*Data availability.* The full MERRA-2 reanalysis dataset including aerosol fields is currently publicly available online through the Goddard Earth Sciences Data and Information Services Center (GES DISC; http://disc.sci.gsfc.nasa.gov/mdisc/). The AERONET and MODIS AOD data can be access at https://aeronet.gsfc.nasa.gov/ and https://modis.gsfc.nasa.gov/, respectively.

*Competing interests.* The authors declare that they have no conflict of interest.

*Acknowledgements.* The first author wishes to thank CNPq (Conselho Nacional de Desenvolvimento Científico e Tecnológico) and CAPES (Coordenação de Aperfeiçoamento de Pessoal de Nível Superior) for the financial support (process 141329/2016-5 and 145857/2017-00,



1  respectively). Additional acknowledge for the group at the Global Modeling and Assimilation Office who produced MERRA-2 and all the

2  persons from AERONET network and MODIS instrument teams for making available the aerosol products used in this study.



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
