# Peer review of "Estimates of direct radiative forcing due to aerosols from the MERRA-2 reanalysis over the Amazon region"

_Atmospheric Chemistry and Physics, 2018_

## Referee Comment (RC1) · S. Kinne (Referee) · 19 Jun 2018

Review of paper:

**Estimates of direct radiative forcing due to aerosol from the MERRA assimilation over the Amazonas region.**

*by B.Penna et al.*

**Positives**

- exploring the seasonality and their inter-annual variations of aerosol direct radiative effects over the Amazon region

**Concerns**

- method-1 results are unlikely to yield reliable aerosol rad.effects other than at the surface
- it is often unclear to what spatial and temporal scales the given variability refers to
- the climate impact relevance is limited

**General comments:**

Estimates for aerosol direct effects at clear-sky conditions are presented for the Amazonas region on different spatial and temporal scales. The highlight of the paper is the analysis of regional output from dual calls (with and without aerosol) of a multi-year MERRA-2 reanalysis and associated GOCART radiative transfer simulations. The results display the strong inter-annual variability of aerosol loading (and associated aerosol direct radiative effects) during the dry season with maxima usually during September. Hereby the confirmation of the retrieved regional AOD variability with MODIS retrievals could be extended to regional AOD retrieval variations of other satellite sensors, such as MISR, SeaWiFS and ATSR.

With the strong seasonal and inter-annual variability all aerosol direct radiative effects are usually offered via large ranges (for TOA, atmosphere and surface). As the associated spatial and temporal scales are often not mentioned, the usefulness of this presentation is somewhat limited, also as no central a multi-year regional annual and multi-year seasonal averages are offered.

The method 1 has deficiencies. The method is only a function of AOD to determine clear-sky radiative effects. Such an approximation may work possibly at the surface, but is unlikely to provide useful results for aerosol direct radiative effects at TOA and thus also for the atmosphere. Aerosol absorption and surface reflectance bring in additional dependencies to be considered. Moreover, the comparison the method 2 is limited, as method 1 apparently applies a non-zero background and also as likely only clear-sky solar radiative effects are addressed in method 1..

And then there is a limited climate relevance as carbon aerosol has relatively as carbon aerosol (as dominant during the biomass burning dry season) has an almost neutral response at the TOA, especially when considering the presence of clouds (which are ignored in this study).

The paper is a nice analysis of long-term MERRA-2 simulations on variability of aerosol and its associated direct radiative effects. Since there are also other parameters which influence the aerosol direct radiative effects it might also be useful to diagnose from the MERRA-2 simulations data for AAOD, possible even hereby separating AOD and AAOD by accumulation mode and coarse mode for aerosol information (in the reanalysis) on both size and absorption. Another parameter to diagnose would be the solar surface albedo. AS another suggestion I would tone down results for method 1 and at best present its estimates for radiative effects at the surface.

**Minor comments:**

The title should be more accurate. The term 'forcing' implies anthropogenic climate relevant radiative effects at TOA and all-sky conditions (with clouds). But here (1) only total aerosol (and

not anthropogenic aerosol) radiative effects and (2) only at clear-sky conditions and (3) in method1 probably only effects on solar radiation are examined. Maybe "**Estimates of aerosol direct radiative effects and their temporal variability over the Amazon region during the last decades based on MERRA-2 reanalysis output**"

1/5 … monthly average **solar** radiative **effects**  at the top …

1/6  the very large values (-10 at TOA and -30 at surf) were only reached in the center of biomass burning …   It is unclear if you talk about inter-annual variability or spatial variability and then again in your paper you show define two different size Amazonas region (your official definition in the text and Merra-3 plots… so 'averages' cannot be really compared)

1/7 it is unclear how natural aerosols are defined …. certainly not by monthly with relatively little aerosol (as there are even natural contributions – e.g. by lightening – during the dry season)

2/19 by the way the contribution from biomass burning to net-flux changes at the TOA (the climate impact) is relatively small with the relative strong absorption, though radiative effects to atmospheric dynamics (solar heating) and surface processes (reduced insolation) certainly matter

2/7 Ok with this regional definition but then show this region via a frame in Figures 6, 7 and 8.

2/16-18 incorrect formulas (the Angstrom parameter is neg. slope in log-space): in the 2.formula a minus sign is missing: alfa = - ln(AOD1/AOD2) / ln(lambda1/lambda2) and 1.formula should be: AOD,550 = exp (-alfa*ln(550/lambda) +ln(AOD,lamda))

4/16 add 'solar'

4/22 I doubt that method 1 can be applied to the region without significant error. It is not clear to me based on what data these polynomial were derived and how they were validated (just for a specific site/time?). When the aerosol absorption changes over time these relationships will likely break down, even if absorption changes are implicitly included (e.g. larger AOD may correlate in that region with larger absorption). I also question the usefulness of such polynomial fits with sign-changing (positive and negative) pre-factors, which yields partial cancellations and does not seem very useful for conceptual understanding which such simple fit should offer. I would have tried a much simple linear fit first, possibly as a function of AAOD as well.

4/26 using AOD data of the wet season as natural reference is poor, especially if absorption (less) and particle size (larger) differ.

5/13  so the MERRA output was diagnosed and hereby SW and LW aerosol direct clear-sky radiative effects were considered .Is this consistent with method 1 or does method 1 just address solar radiations impacts (which then should be ca 10%  stronger)?)

8/8 It is unclear how the forcing is determine with method 1, as there are dependencies on surface albedo and absorption which cannot be consider by the simplicity of method 1

9/1 apparently the forcing variability that was mentioned in the abstract refers to the interannual variability during the dry season .. that should be made clear in the abstract.

10/2 … because higher AOD values in that region are stronger absorbing so, and with stronger absorption the direct aerosol efficiency decreases. (Still, here just clear-sky TOA effects are addressed but all-sky TOA effects are much more interesting). Also the TOA effect depends on variations to the solar surface albedo.

11/8  It also would be great not just to address regional range but actually determine multi-annual averages as typical value, possibly with a seasonal and monthly dependence (which then could be compared to the MACv2 climatology data – see below).

14/9  It was not clear that the method 1 fit only refers to dry-season AOD minus wet season AOD data. This even further reduces potential comparisons. Clearly this difference should be spelled out much clearer before comparing rad. forcing results. Anyway for me such as simple method 1 AOD approach has only (limited merit) for impact on surface (net-) fluxes, and results for TOA and thus also the atmosphere should be marked as highly unreliable – and I would remove those results from the paper.

below are for reference results (and comparisons) of off-line simulations for aerosol direct radiative effects with the MACv2 aerosol (monthly) climatology for the Amazonas region:

1. TOA and surface effects - annual
2. cloud effects and atmospheric effects - annual
3. clear-sky effects – seasonal
4. surface and atmos clear-sky effects - monthly

[Figure]

**Figure 1** *Annual maps for radiative effects of today's average aerosol (left), its solar effect only (center) and the anthropogenic impact. Clear sky effects are presented in the upper two rows: at the TOA (row 1) and at the surface (row2) and all-sky effects (with ISCCP clouds) are presented in the lower two row: at the TOA (row 3) and at the surface (row 4). Note, data in the annual map for today's anthropogenic aerosol direct forcing (column 3, row3) are multiplied by a factor 10. Blue colors indicate a cooling and red colors a warming. Values below the label indicate regional averages.*

| | | |
|---|---|---|
| clear-sky TOA  (solar +IR) | -2.7W/m2 | |
| clear-sky surface (solar +IR) | -9.7W/m2 | |
| **clear-sky TOA (solar)** | **-3.5W/m2** | ( -1.0W/m2 anthrop ) |
| **clear-sky surface (solar)** | **-11.0W/m2** | ( -3.5W/m2 anthrop ) |

| | | |
|---|---|---|
| all-sky TOA  (solar +IR) | -1.4 W/m2 | |
| all-sky surface (solar +IR) | -8.8 W/m2 | |
| all-sky TOA (solar) | -1.8 W/m2 | (-0.54W/m2 anthrop) |
| all-sky surface (solar) | -9.7 W/m2 | (-3.1 W/m2 anthrop) |

[Figure]

**Figure 2** *Annual maps for cloud effects ('all-sky' minus 'clear-sky': top two rows) and of aerosol atmospheric effects ('TOA' minus 'surface': bottom two rows) for today's average aerosol (left), its solar effect only (center) and the anthropogenic impact. Cloud effects are presented at TOA (row1) and at the surface (row2). Note due to the aerosol solar absorption the relative reduction s at the TOA are much larger than at the surface. Aerosol atmospheric effects are presented at clear-sky (row 3) and all-sky (with ISCCP clouds) conditions (row 4). The positive cloud-effect values show that clouds cause the direct forcing to be less negative and the positive (red to green colors) indicate that major contributions are by solar absorption.*

| clear-sky absorption (solar+IR) | +7.0 W/m2 |
|---|---|
| all-sky absorption (solar +IR) | +7.4 W/m2 |

| **clear-sky absorption (solar):** | **+8.0 W/m2** |
|---|---|
| all-sky absorption (solar): | +7.9 W/m2 |

[Figure]

**Figure 3** *seasonal clear-sky aerosol radiative effects over the MERRA Amazonas region.*

[Figure]

***Figure 4*** *monthly clear-sky aerosol rad. effects at surf and atm over MERRA Amazonas region.*

---

## Referee Comment (RC2) · Anonymous Referee #2 · 20 Jun 2018

This study presents clear-sky direct aerosol radiation forcing results for a period of 16 years over the Amazon area. The scientific hypothesis is based on the comparison between MERRA-2 reanalysis data with and without aerosol loading and using two different methods to obtain the direct aerosol radiative forcing (M1 and M2). Although this is an interesting topic I suggest that this manuscript cannot be accepted in ACP at this stage. My main concerns are: (1) It is not clear what is the added value of using method 1. The authors should either elaborate more to show if there is any scientific significance in including M1 results or they should eliminate these results from their study. (2) The authors should provide physically based explanations on the large deviations between different datasets and methods (e.g. in Table 2). (3)

The authors should state clearly and justify if there is any improvement in aerosol radiative forcing constraints compared to previous relevant studies. (4) The cloud-free radiation estimations presented here are most probably irrelevant when it comes to actual climate change considerations.
* * *

---

## Referee Comment (RC3) · Anonymous Referee #3 · 20 Jun 2018

This study presents an analysis of the aerosol optical depth (AOD) and the clear-sky aerosol direct radiative effect (DRE) based on observations from ground-based (AERONET) and satellite sensors (MODIS) and from the recent NASA MERRA-2 re-analysis, which assimilates AOD from numerous NASA satellites and AERONET.

While the paper presents a nice analysis of the seasonality of AOD from both observations and analysis, there are a number of weaknesses in its analysis of the DRE. First, the empirical "Method 1" for calculating DOE parameterized only by AOD is not justified sufficiently; while the cited work may have found this relationship to work for their considered purposes, there seems no reason for it to work all the time, particularly

at the top of the atmosphere where the DRE depends on other aerosol parameters (e.g. single scattering albedo) and surface albedo (see Chylek and Wong, GRL, 1995). Furthermore, "Method 1" and "Method 2" (which is really just the output of the double radiation call from MERRA-2) are not comparable because (a) Method 1 assumes a baseline AOD = 0.11 while Method 2 has a clean baseline (AOD = 0.0) and (b) Method 1 is applicable for a single wavelength while Method 2 is for broadband conditions in the shortwave and longwave.

I recommend that this study eliminate consideration of Method 1 and refocus their efforts on evaluating the MERRA-2 aerosol products over the Amazon region. As an example, they can see the case studies of MERRA-2 evaluation presented in Buchard et al. 2017 (J. Clim). There are numerous aerosol products available from MERRA-2 that can be examined further over this region using a number of observational datasets. Such an analysis will provide stronger evidence of the authors' suggestion that MERRA-2 is a useful tool for examining the DRE over this important biomass burning region.

Minor comments are below:

Page 1, Line 25: "…and the total amount of aerosols is generally obtained …". Do you mean "ARF is generally calculated for all aerosols combined, as estimates by species are less consistent". Also, ARF should be DRE - this is not a forcing because it does not reference some past state, and because natural aerosols are included.

Page 2, Line 14. It should not be "assimilation of aerosols" but "assimilation of aerosol optical depth"

Page 2, Line 17-18. "Thus, MERRA-2 is a great tool for studies of aerosols and their impact on climate". I would not say this is a given. Remove this sentence.

Page 3, Lines 16-19. Remove Lines 16-19 and the equations. It is sufficient to state "The AOD of 550 nm was obtained using the Angstrom relationship. Also, the equations

are incorrect.

Remove Page 3, line 25. MODIS AOD products are available at 0.47, 0.55, and 0.65 microns; be careful not to imply they are available throughout the wide spectral range.

Page 4, Line 10. MODIS Collection 5 Neural Net Retrieval Equation 4: Is this an AOD at a specific wavelength? What wavelength?

Page 5, Lines 1-13. This whole section can be shortened to simply: We calculate the DRE using output from the MERRA-2 double radiation call for clear (no clouds) and clean and clear (no clouds or aerosols) conditions in both the long wave and short-wave. Diagnostics are available from the tavgM_2d_rad_Nx collection (provide the doi reference for this collection).

Page 5, Line 22: Randles et al. 2017 provided comparisons to AERONET at Alta Floresta; how do your results compare?

Page 7, Table 1: Do you sample the reanalysis like MODIS or is this just a comparison of monthly means? MODIS has a clear-sky bias and this can impact comparisons made in such a fashion
* * *